# Novel Human/Non-Human Primate Cross-Reactive Anti-Transferrin Receptor Nanobodies for Brain Delivery of Biologics

**DOI:** 10.3390/pharmaceutics15061748

**Published:** 2023-06-16

**Authors:** Laura Rué, Tom Jaspers, Isabelle M. S. Degors, Sam Noppen, Dominique Schols, Bart De Strooper, Maarten Dewilde

**Affiliations:** 1Laboratory for Therapeutic and Diagnostic Antibodies, Department of Pharmaceutical and Pharmacological Sciences, KU Leuven, 3000 Leuven, Belgium; 2VIB-KU Leuven Center for Brain & Disease Research, 3000 Leuven, Belgium; 3Laboratory for the Research of Neurodegenerative Diseases, Department of Neurosciences, Leuven Brain Institute (LBI), KU Leuven, 3000 Leuven, Belgium; 4Laboratory of Virology and Chemotherapy, Department of Microbiology, Immunology and Transplantation, Rega Institute, KU Leuven, 3000 Leuven, Belgium; 5UK Dementia Research Institute, University College London, London WC1E 6BT, UK

**Keywords:** nanobody, VHH, transferrin receptor, blood-brain barrier, receptor-mediated transcytosis

## Abstract

The blood-brain barrier (BBB), while being the gatekeeper of the central nervous system (CNS), is a bottleneck for the treatment of neurological diseases. Unfortunately, most of the biologicals do not reach their brain targets in sufficient quantities. The antibody targeting of receptor-mediated transcytosis (RMT) receptors is an exploited mechanism that increases brain permeability. We previously discovered an anti-human transferrin receptor (TfR) nanobody that could efficiently deliver a therapeutic moiety across the BBB. Despite the high homology between human and cynomolgus TfR, the nanobody was unable to bind the non-human primate receptor. Here we report the discovery of two nanobodies that were able to bind human and cynomolgus TfR, making these nanobodies more clinically relevant. Whereas nanobody BBB00515 bound cynomolgus TfR with 18 times more affinity than it did human TfR, nanobody BBB00533 bound human and cynomolgus TfR with similar affinities. When fused with an anti-beta-site amyloid precursor protein cleaving enzyme (BACE1) antibody (1A11AM), each of the nanobodies was able to increase its brain permeability after peripheral injection. A 40% reduction of brain Aβ_1–40_ levels could be observed in mice injected with anti-TfR/BACE1 bispecific antibodies when compared to vehicle-injected mice. In summary, we found two nanobodies that could bind both human and cynomolgus TfR with the potential to be used clinically to increase the brain permeability of therapeutic biologicals.

## 1. Introduction

The blood-brain barrier (BBB) is a very specialized organ that, together with the other brain barriers such as the blood-cerebrospinal fluid barrier and blood-arachnoid barrier, contributes to the isolation of the central nervous system (CNS) from the rest of the organism. This ensures that harmful circulating substances in the peripheral blood flow do not freely reach the CNS, while still allowing a selective influx of required elements such as nutrients [1,2]. Hence the BBB represents a bottleneck for the treatment of neurological diseases, as most of the biologicals are not able to reach their brain targets or only do so in very small quantities, i.e., less than 0.1% of peripherally administered doses [3,4,5]. Therefore, the high doses that need to be administered result in potential side effects and high treatment costs. The BBB is composed of an endothelial layer which is surrounded by pericytes and astrocyte end-feet [1]. Tight junctions expressed by the BBB endothelium limit the paracellular diffusion of substances. Most of the required substances in the brain that come from the periphery follow an active route of entry via specific channels and transporters [6]. Receptor-mediated transcytosis (RMT) is one such physiological mechanism in which nutrients are recognized by specific receptors that are expressed on the surface of the endothelial cells, internalized in intracellular vesicles, and finally released in the brain parenchyma [7]. Targeting such RMT receptors with antibodies is a valid strategy to increase the brain permeabilities of biologicals [8]. Among these receptors, the transferrin receptor (TfR) is one of the most exploited RMT mechanisms for brain drug delivery [9,10,11,12,13,14,15,16,17,18]. Recently, an anti-TfR-idursulfase conjugate drug (Izcargo^®^) was approved in Japan for the treatment of Hunter syndrome [19].

Nanobodies, which are the variable domain of camelid heavy chain-only antibodies, have ideal properties that allow the engineering of multispecific constructs [20]. Several nanobodies that successfully deliver biologicals over the BBB by targeting RMT receptors such as TfR, IGF1R, or TMEM30A have been described [11,15,16,21,22]. In a previous work, we obtained a set of mouse TfR binders and a set of human TfR binders [15,16]. Unfortunately, our human TfR nanobodies did not bind cynomolgus TfR despite the high sequence homology between both proteins. Lack of binding to non-human primate (NHP) TfR represents an obstacle to determine the preclinical efficacy and safety of potential therapeutic conjugates. Regulatory guidelines indicate that two animal species should generally be used for non-clinical toxicity testing, thereby supporting the development of drugs for human use: a rodent (e.g., mouse or rat) and a non-rodent (e.g., dog or NHP) [23]. Here we describe the identification of two human/cynomolgus TfR-binding nanobodies and validate in vivo their potential to shuttle therapeutics into the brain.

## 2. Materials and Methods

### 2.1. Animals

All animal experiments were conducted in compliance with the commonly accepted ‘3Rs’, according to protocols approved by the local Ethical Committee of Laboratory Animals at KU Leuven (governmental license LA1210579, ECD project number P213/2020), and following governmental and EU guidelines. Mice were housed under standard conditions according to the guidelines of KU Leuven, with a 12-h light-dark cycle and with access to food and water ad libitum. Humanized Tfrc mice (hAPI KI mice), which express a chimeric mouse TfR with the human apical domain under the endogenous promoter, were used for this study [16].

### 2.2. Immunization and Nanobody Library Preparation

Targeted nanobody libraries were obtained in collaboration with the VIB Nanobody Core (Brussles, Belgium). Three alpacas were subjected to four biweekly DNA immunizations using recombinant pVAX1 plasmid DNA (Thermo Fisher Scientific, Waltham, MA, USA) encoding for a chimeric alpaca TfR with the cynomolgus apical domain (synthesized at Twist Biosciences, South San Francisco, CA, USA). DNA solutions were injected intradermally at multiple sites on the front and back limbs near the draining lymph nodes, and this was followed by electroporation. On days 4 and 8, after the last immunization, blood samples were collected and pooled, and total RNA from peripheral blood lymphocytes was isolated to recover the nanobody-encoding genes. The phagemid library was prepared as previously prescribed [24]. Briefly, total RNA was used as a template for first-strand cDNA synthesis with oligodT primer. This cDNA was used to amplify the nanobody-encoding open reading frames by means of polymerase chain reaction (PCR), digested with *PstI* and *NotI*, and cloned into a phagemid vector (pBDS001, a modified pMECS vector with an insertion of 3xFlag/6xHis tag at the C-terminus of the nanobody insertion site). Electrocompetent *E. coli* TG1 cells (Biosearch Technologies, Middlesex, UK) were transformed to obtain the nanobody libraries.

### 2.3. Cell Line Generation

The Flp-In™-CHO™ system (Thermo Fisher Scientific) was used to generate stable Chinese hamster ovary (CHO) cell lines overexpressing cynomolgus or human TfR. DNA encoding for the cynomolgus TfR or the human TfR, tagged with hemagglutinin (HA) at the C-terminus and followed by green fluorescent protein (GFP) under the control of an internal ribosome entry site (IRES), was synthesized and subcloned by Twist Bioscience into the pcDNA™5/FRT mammalian expression vector (Thermo Fisher Scientific). Flp-In™-CHO™ cells were maintained with Gibco™ Ham’s F-12 Nutrient Mix medium supplemented with GlutaMAX™ (Thermo Fisher Scientific), 10% fetal bovine serum (FBS), and 100 μg/mL Zeocin™ selection antibiotic (Invivogen, San Diego, CA, USA) until the day of transfection. Cells were transfected with the TransIT-PRO^®^ Transfection kit (Mirus, Madison, WI, USA) and maintained in Gibco™ Ham’s F-12 Nutrient Mix medium supplemented with GlutaMAX™ (Thermo Fisher Scientific), 10% FBS, and Hygromycin B Gold (Invivogen) to select for stable transfectants. Stable transfectants were then amplified and frozen with 10% dimethyl sulfoxide (DMSO) for further use.

### 2.4. Nanobody Selection, Expression and Purification

Nanobody-displaying M13 phage libraries were prepared according to standard protocols [24] and selected twice on TfR-overexpressing cells. Briefly, 6 × 10^11^ cfu of phages were blocked with phosphate-buffered saline (PBS)/10% FBS and incubated for an hour with 5 million cells of CHO-cynomolgus TfR-overexpressing cells in the first selection round and CHO-human TfR-overexpressing cells in the second selection round. Non-binding phages were discarded with 5 consecutive washing steps with PBS/10% FBS, whereas bound phages were eluted by means of trypsinization. The output phage library of the second selection round was subcloned into an expression vector (pBDS119, a modified pHEN6 vector with an OmpA signal peptide and a C-terminal 3xFlag/6xHis tag) and transformed in *E. coli* TG1 cells. Single clones were picked, sequenced (Eurofins, Luxembourg), and clustered according to sequence homology with PipeBio (Horsens, Denmark). In addition, the small-scale expression of sequenced clones was performed and periplasmic extracts were prepared as previously described [24] and screened for direct binding to CHO-human TfR-overexpressing cells. Nanobody leads were expressed and purified according to the protocol provided by Pardon et al. [24].

### 2.5. Flow Cytometry-Based Binder Screening and Validation

Periplasmic extracts that were diluted 1:10 in PBS 2% FBS, or a dilution range of different nanobody or bispecific antibody concentrations prepared in PBS 2% FBS, were incubated with 0.1 million CHO cells overexpressing either the human, cynomolgus, or mouse TfR for 30 min at 4 °C. As control for background binding, periplasmic extracts, nanobodies, or bispecific antibodies were also incubated with 0.1 million CHO cells overexpressing GFP. The binding of nanobodies was next resolved by a 30-min incubation at 4 °C with an anti-FLAG-iFluor647 antibody (A01811, Genscript, Piscataway, NJ, USA), diluted 1:500 for screening and 1:250 for validation assays, or with anti-human IgG Fc-Alexa Fluor647 antibody (410714, BioLegend, San Diego, CA, USA) diluted 1:200. Dead cells were stained with the viability dye eFluor™780 (1:2000; 65-0865-14, Thermo Fisher Scientific) for 30 min at 4 °C. Cells were fixed with 4% paraformaldehyde before being analyzed. Flp-In™-CHO™ cells, used as unstained control and single stain controls, were used to determine the cutoff point between background fluorescence and positive populations. UltraComp eBeads™ Compensation Beads were used (Thermo Fisher Scientific) to generate single stain controls of both anti-FLAG-iFluor647 antibody and anti-human IgG Fc-Alexa Fluor647 antibody. The data was acquired by using an Attune Nxt flow cytometer (Invitrogen) and analyzed by FCS Express 7 Research Edition.

### 2.6. Surface Plasmon Resonance

Surface Plasmon Resonance (SPR) was used to measure the interactions between nanobodies and recombinant human or cynomolgus TfR receptor. Human TfR (2474-TR, R&D Systems, Minneapolis, MN, USA) and cynomolgus TfR (90253-C07H, Sino Biological, Beijing, China) were biotinylated with the EZ-Link NHS-PEG4-Biotinylation Kit (ThermoFischer Scientific) according to the manufacturer’s instructions. The binding experiments were performed at 25 °C on a Biacore T200 instrument (Cytiva, Marlborough, MA, USA) in HBS-EP+ buffer (10 mM HEPES, 150 mM NaCl, 3 mM EDTA, and 0.05% *v*/*v* Surfactant P20). Biotinylated human TfR and cynomolgus TfR were captured on a streptavidin-coated SA sensor chip (Cytiva) at a density of around 250 RU. Increasing concentrations of nanobodies were sequentially injected in a single cycle at a flow rate of 30 μL/min. The dissociation was monitored for 20 min. No specific regeneration was needed. A reference flow was used as a control for non-specific binding and refractive index changes. Several buffer blanks were used for double referencing. Binding affinities (K_D_) and kinetic rate constants (k_a_, k_d_) were derived after fitting the experimental data to the 1:1 binding model with the Biacore T200 Evaluation Software 3.1 using the single-cycle kinetic procedure. Each interaction was repeated at least three times.

### 2.7. Bio-Layer Interferometry (BLI)

Binding of the bispecific antibodies to beta-site amyloid precursor protein cleaving enzyme (BACE1) was assessed with an Octet RED96 (Sartorius, Göttingen, Germany). Briefly, streptavidin (SA) biosensors (18-5020, Forté Bio/Molecular Devices) were pre-wet for at least 10 min in kinetic buffer. Next, the biosensors were dipped in biotinylated BACE1 (5 µg/mL in kinetic buffer). BACE1 (Protein Service Facility, VIB) biotinylation was performed with the EZ-Link NHS-PEG4-Biotinylation Kit (ThermoFischer Scientific) according to the manufacturer’s instructions. Biosensors were then sequentially submerged in baseline wells filled with PBS containing 0.02% Tween, 0.1% bovine serum albumin (BSA), and bispecific antibodies diluted in the same buffer, and finally back into baseline wells to assess dissociation. Data was recorded using the Forté Bio Octet RED data acquisition software (Forté Bio/Molecular Devices). Curve fitting and binding kinetics determination was performed with a 1:1 model interaction using the Forté Bio Octet RED analysis software (Forté Bio/Molecular Devices). Sensorgrams were generated using Graphpad.

### 2.8. Bispecific Antibodies Engineering and Expression

The anti-BACE1 affinity-matured 1A11 antibody (1A11AM) was used to generate bispecific antibodies. The engineering and expression of BBB00574 (1A11AM-BBB00515) and BBB00578 (1A11AM-BBB00533) were performed as previously described [16,25]. Briefly, the DNA sequences encoding for the heavy chain composed of BBB00515 or BBB00533 fused to an Fc with knobs-into-holes (KiH) and ablated effector function mutations (human IgG1, L234A, L235A, P329G, T350V, T366L, K392L, T394W), the 1A11AM heavy chain with KiH and ablated effector function mutations (human IgG1, L234A, L235A, P329G, T350V, L351Y, F405A, Y407V), and the 1A11AM light chain (human kappa) were synthesized by Twist Bioscience and cloned in their pTwist CMV BetaGlobin WPRE Neo vector (Twist Bioscience). Expressions were performed using the Mirus CHOgro^®^ High Yield Expression System (Mirus Bio) according to the manufacturer instructions. Briefly, ExpiCHO-S™ cells (Mirus Bio) were transfected with the nanobody-human Fc fusions and the 1A11AM heavy chain and light chain with a transfection ratio of 2:1:3 with TransIT-PRO Transfection Reagent (Mirus Bio) [25]. After an incubation time of 14 days, the antibodies were purified, first with AmMag™ Protein A Magnetic Beads (Genscript) and then over a CaptureSelect™ CH1-XL Pre-packed Column (ThermoFischer Scientific) according to the manufacturer’s instructions.

### 2.9. Sample Collection, Aβ Extraction and Enzyme-Linked Immunosorbent Assay (ELISA)

Each mouse was euthanized by the intraperitoneal injection of a Dolethal overdose (150–200 mg/kg). To harvest plasma, blood was collected with a prefilled heparin syringe via cardiac puncture. Next, blood samples were spun at 2000× *g* for 10 min and plasma was collected. Brains were harvested after transcardial perfusion with heparinized PBS. Mouse Aβ_1–40_ samples from brain and plasma were prepared according to the protocols used by Serneels et al. [26]. Briefly, one brain hemisphere per mouse was homogenized in buffer containing 0.4% diethylamine (Sigma, St. Louis, MO, USA) and 50 mM NaCl supplemented with cOmplete™ protease inhibitor cocktail (Roche, Basel, Switzerland) using a FastPrep-24™ Classic bead-beating lysis system (MP Biomedicals, Santa Ana, CA, USA). Next, soluble Aβ_1–40_ was extracted via 0.4% diethylamine treatment for 30 min at 4 °C, high-speed centrifugation (100,000× *g*, 1 h, 4 °C), and neutralization with 0.5 M Tris-HCl (pH 6.8). Soluble Aβ_1–40_ levels extracted from brain and Aβ_1–40_ levels in plasma were quantified by ELISA using Meso Scale Discovery (MSD) 96-well plates and Aβ_1–40_ antibodies provided by Janssen Pharmaceutica (Beerse, Belgium). JRFcAβ40/28 antibody was used as the capture antibody whereas the rodent-specific JRF/rAβ/2, labeled with sulfo-TAG, was used as the detection antibody. Soluble Aβ_1–40_ levels were interpolated from a recombinant Aβ_1–40_ (A-1153-2, rPeptide) standard curve with the non-linear regression fit Log (agonist) vs. response–variable slope (4 parameters) model from the Graphpad prism 9.4.1 software.

### 2.10. Statistical Analysis

Statistics were performed using the Graphpad prism 9.4.1 software. A one-way ANOVA statistical test, followed by a Dunnett’s multiple comparisons test, was performed to report statistically significant differences in the levels of Aβ_1–40_ in the plasma and brain samples obtained from mice injected with PBS or bispecific antibodies. Statistical significance was considered for a *p* value < 0.05.

## 3. Results

### 3.1. Identification of Human/Cynomolgus TfR Binders

The currently described TfR affinity binders that are able to cross the BBB bind the TfR apical domain. To direct immunization against this region, camelids were immunized with DNA encoding for alpaca TfR with the sequence encoding for the apical domain replaced by the cynomolgus apical domain sequence. Phage libraries, generated after 4 biweekly immunizations of two different camelids, underwent two rounds of pannings on CHO cell lines, overexpressing cynomolgus TfR in the first round and human TfR in the second round (Figure 1A). A total of 95 individual clones were then picked from the resulting selected library and sequenced (Figure 1A). These nanobody sequences were clustered together to exclude identical sequences. The periplasmic extracts of all the non-redundant sequences were then prepared and screened to find binders to both human and cynomolgus TfR-overexpressing CHO cells (Figure 1A). From the 32 clones screened, only 8 bound both human and cynomolgus TfR-overexpressing CHO cells. A cluster analysis of these 8 clones revealed the high homology (above 91%) between 7 of them. Therefore, out of these 8 hits, two leads were chosen for further expression, purification, and characterization. Purified BBB00515 and BBB00533 bound both human and cynomolgus TfR-overexpressing cells whereas they did not bind mouse TfR-overexpressing cells or a control cell line only expressing GFP (Figure 1B–E). The binding kinetics to recombinant human and cynomolgus TfR were further characterized in vitro with Surface Plasmon Resonance (SPR) (Figure 1F–H). Both BBB00515 and BBB00533 bound immobilized recombinant cynomolgus TfR with similar estimated affinity constants (K_D_ = 63.00 ± 1.20 nM for BBB00515 and K_D_ = 103.77 ± 8.14 nM for BBB00533; Figure 1H). Both also bound to immobilized recombinant human TfR, but with higher K_D_ values (K_D_ = 1183.67 ± 423.81 nM for BBB00515 and K_D_ = 207.00 ± 27.84 nM for BBB00533; Figure 1H).

### 3.2. Anti-Human/Cynomolgus TfR Nanobodies Shuttle Anti-BACE1 mAb into the Brain

BACE1 inhibition in the brain was the paradigm used to assess the potential of the nanobodies to deliver, in vivo, biologicals in the brain [12,13,16]. BACE1 is responsible for the cleavage of APP to give rise to Aβ species [27]. BACE1-inhibiting antibody (Mab 1A11AM) is able to reduce brain Aβ_1–40_ levels in vivo but does not cross the BBB in sufficient amounts when it is delivered peripherally [16,28]. Here, the abilities of BBB00515 and BBB00533 to modulate the properties of 1A11AM and improve brain accumulation after peripheral delivery were examined. Therefore, bispecific antibodies with ablated effector function were generated, with each bispecific antibody having one intact 1A11AM Fab and one nanobody, both fused to their respective Fcs (Figure 2A). BBB00574 bispecific antibody carries a BBB00515 nanobody, whereas BBB00578 carries a BBB00533 nanobody. As expected, both bispecific antibodies were still able to bind hTfR in living cells, but were not able to do so in a negative-control cell line (Figure 2B,C). Binding to BACE1 was confirmed with BLI, in which biotinylated recombinant human BACE1 protein was immobilized at the tip of streptavidin-coated biosensors (Figure 2D–F). Both bispecific antibodies bound human BACE1 with similar K_D_ values of 0.3 nM (Figure 2F).

Next, both bispecific antibodies were administered intravenously in hAPI KI mice in which the mouse TfR apical domain had been replaced by the human sequence [16]. The chosen concentration to inject was 167 nmols/kg, which was the dose at which no central BACE1 inhibition had been observed for 1A11AM after intravenous injection [16,28]. Plasma and brains were harvested 24 h later, and Aβ_1–40_ levels were quantified with ELISA. Both BBB00574 and BBB00578 bispecific antibodies could lower Aβ_1–40_ levels in plasma by 60%, as compared to what had been found in samples of PBS injected mice (Figure 3A). Interestingly, Aβ_1–40_ levels in the brain were reduced by 40% as compared to what had been noted in PBS injected mice (Figure 3B), suggesting the ability of both nanobodies to carry biological moieties over the BBB. 1A11AM antibody coupled to an anti-GFP nanobody and administered intravenously at the same concentration of 167 nmols/kg in hAPI KI mice did not decrease brain Aβ_1–40_ levels 24 h after injection as compared to what had been noted in a PBS control group [16].

## 4. Discussion

In previous studies, we reported the discovery of first an anti-mouse TfR nanobody (NB62) and next an anti-human TfR nanobody (NB188) [15,16]. These could both efficiently deliver therapeutic moieties in the brain. Unfortunately, we later discovered a lack of NHP cross-reactivity for NB188, and this was a drawback for preclinical development, as safety of potential therapeutics is commonly assessed in NHPs [23]. Previous experiences from our lab and others indicate that acquiring human and cynomolgus TfR binders is challenging [12]. Currently described TfR affinity binders that are able to reach the CNS bind the TfR apical domain. Despite the fact that the apical domains of human and cynomolgus TfR have a 95% homology, there is no guarantee of cross-reactivity of antibodies [12]. We have also observed that two residues in the cynomolgus sequence that differ from those in the human sequence give rise to a glycosylation site that is absent in the human TfR, potentially hampering the binding of biologicals to TfR. To drive immunization against the cynomolgus apical domain, camelids were immunized with a chimeric DNA encoding for the alpaca TfR sequence with the apical domain sequence replaced by the cynomolgus sequence. After phage selections and screening on TfR-overexpressing cells, we found 2 nanobodies, BBB00515 and BBB00533, that were able to bind both human and cynomolgus TfR. Ideally, to be able to predict—based on the outcomes of preclinical studies—the amount of IgG that will accumulate in the brain in human patients, the binding affinities to both human and cynomolgus TfR should be similar. Although some studies have reported TfR binders with similar affinities to human and cynomolgus species [10,13], some others have reported higher affinities to human TfR than to cynomolgus TfR [12,29]. One of our two newly identified nanobodies, BBB00515, binds cynomolgus TfR with 18 times more affinity than human TfR. However, BBB00533 is only two times more affine to cynomolgus than human TfR, making it the most interesting nanobody to consider for further preclinical development.

Nanobodies were fused to a BACE1-inhibiting antibody, 1A11AM, to validate their potential to deliver biologicals into the brain. Two bispecific antibodies were engineered, and in each of them, 1A11AM Fab and one of the nanobodies (BBB00515 or BBB00533) were fused to an Fc. To provide a functional readout of BACE1 inhibition and hence BBB penetration, we chose here to quantify the Aβ_1–40_ levels 24 h after injection of the bispecific antibodies. We previously demonstrated that while the 1A11AM antibody is able to reduce Aβ_1–40_ levels in the periphery, it is not able to do so in the brain when injected intravenously [16,28]. Other studies showed similar findings when anti-BACE1 antibodies were injected at low concentrations and allowed to circulate for only 24 h [12,30] in each case. Interestingly, our new bispecific antibodies, BBB00574 and BBB00578—carrying either BBB00515 or BBB00533 nanobodies, respectively—were able to reduce Aβ_1–40_ levels in the brain, suggesting penetration into the brain parenchyma after peripheral injection. Despite having different affinities for human TfR, both of our lead nanobodies, when fused to 1A11AM, were able to inhibit BACE1 by similar levels, resulting in a 40% reduction of Aβ_1–40_ levels in the brain as compared to what had been recorded in the PBS-injected controls. This degree of BACE1 inhibition was similar to those of both Genentech’s anti-human TfR/BACE1 antibodies and our previously reported anti-human TfR nanobody fused to 1A11AM antibody, all injected at equimolar concentrations [12,13,16]. Previous findings corroborate that bispecific antibodies with differing human TfR affinities can lead to similar BACE1 inhibitions 24 h after treatment [12,13]. We strongly believe that this lowering in Aβ-levels in the brain is related to an increased brain penetration of the BACE1-inhibiting antibodies and not to peripheral effects such as those suggested by the peripheral sink hypothesis in the context of Aβ clearance. This hypothesis states that Aβ clearance in blood could yield into an efflux of Aβ from the brain to blood, thereby reducing Aβ in the brain compartment [31]. However, such an effect has not been observed so far in a multitude of studies from various research groups when inhibiting peripheral BACE1 either genetically or pharmacologically, even after several months of treatment [12,16,28,30,32]. Additional experiments, such as pharmacokinetic studies of the bispecific antibodies, could potentially be performed to determine exact antibody levels in the brain over time. Finally, it would be of interest to determine the yield of BACE1 inhibition in the brain/cerebrospinal fluid (CSF) in an NHP. We expect our bispecific constructs to be efficient since their affinities to cynomolgus TfR are similar to their affinities to human TfR and in the same range as the anti-TfR1/BACE1 antibody from Genentech, which yielded a 50% BACE1 reduction in NHP CSF [13]. 

## 5. Conclusions

Here we have described two novel nanobodies binding to both human and non-human primate TfR and suggested that they are able to deliver biologicals into the brain in a humanized TfR mouse model. One of them, BBB00533, binds with similar affinity to human and cynomolgus TfRs. This is an important finding since cross-reactivity is a prerequisite for further clinical development. If their potentials to deliver biologicals were to be confirmed in a non-human primate, these nanobodies could be used clinically to increase the brain permeabilities of therapeutic biologicals.

## Figures and Tables

**Figure 1 pharmaceutics-15-01748-f001:**
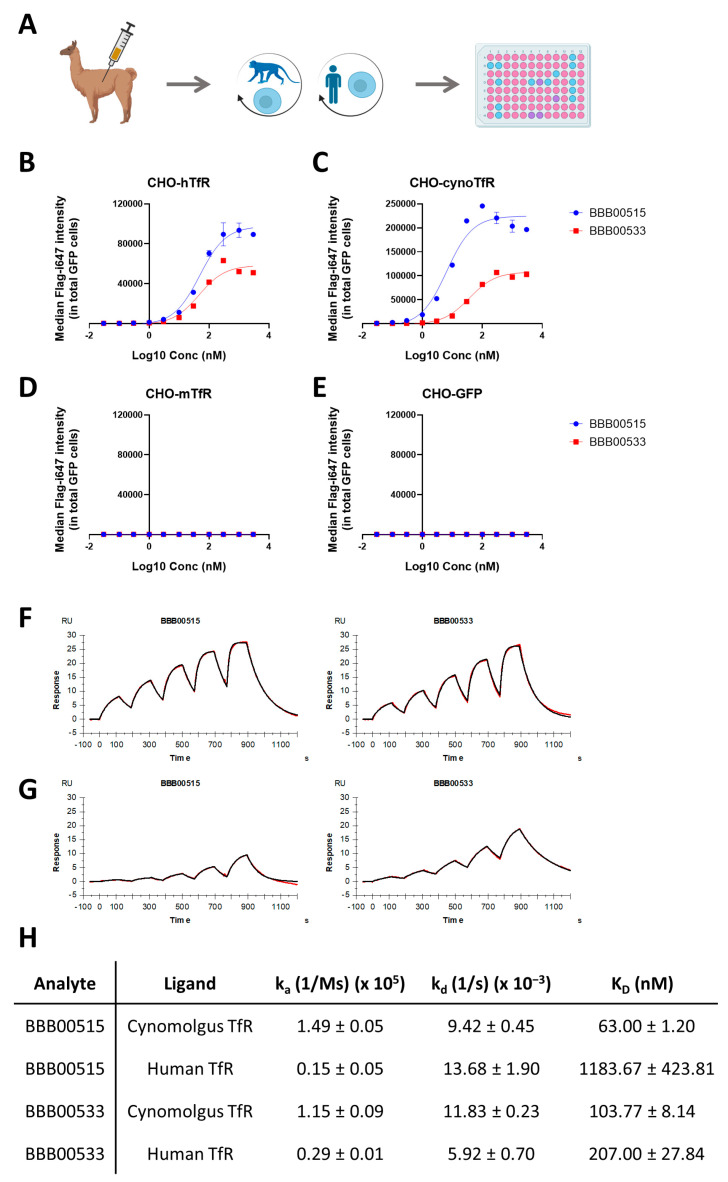
(**A**) Camelids were immunized with alpaca TfR DNA with the human apical domain sequence. Immune phage libraries were selected in the first round on CHO cells overexpressing cynomolgus TfR and on CHO cells overexpressing human TfR in the second round. The sequencing and screening of several clones was performed via flow cytometry based on the binding of periplasmic extract to CHO cells overexpressing human TfR. Graphical designs were created with BioRender.com. Different colors in 96-well plate represent different levels of binding (**B**–**E**) Flow cytometry analysis showing the binding of nanobodies to CHO cells overexpressing (**B**) hTfR, (**C**) cynomolgus TfR, (**D**) mouse TfR, and (**E**) GFP. The data represent means ± SEM (*n* = 3). (**F**,**G**) SPR binding kinetics (black-colored line) and curve fitting (red-colored line) of nanobodies binding to cynomolgus TfR (**F**) and human TfR (**G**) recombinant material. (**H**) SPR kinetic analysis indicated different binding affinities of nanobodies to cynomolgus and human TfR recombinant material.

**Figure 2 pharmaceutics-15-01748-f002:**
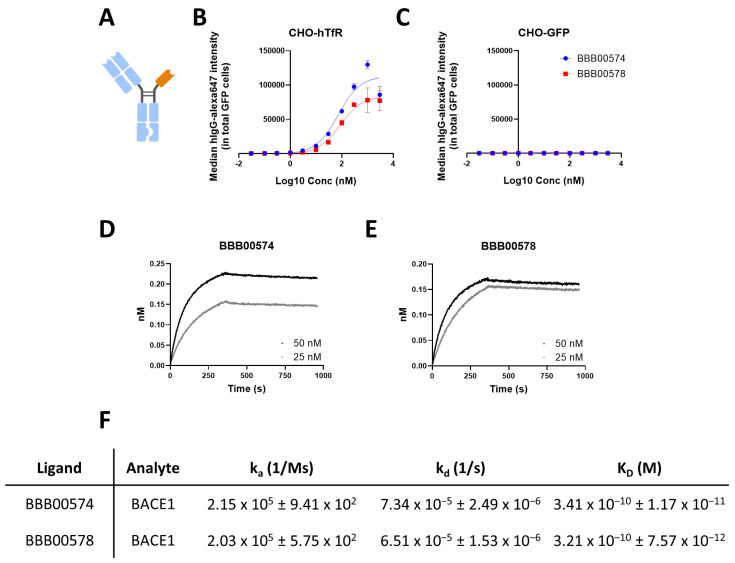
(**A**) A bispecific antibody design with one 1A11AM intact Fab and a nanobody coupled to an Fc with KiH and ablated effector function mutations. Created with BioRender.com. (**B**,**C**) Flow cytometry analysis showed bispecific antibody binding to (**B**) human TfR or (**C**) GFP-overexpressing cells. The data represent means ± SEM (*n* = 3). (**D**,**E**) BLI binding kinetics of BBB00574 (**D**) and BBB00578 (**E**) bispecific antibodies to recombinant human BACE1. (**F**) The kinetic analysis of bispecific antibodies binding human BACE1 was performed using a 1:1 interaction model.

**Figure 3 pharmaceutics-15-01748-f003:**
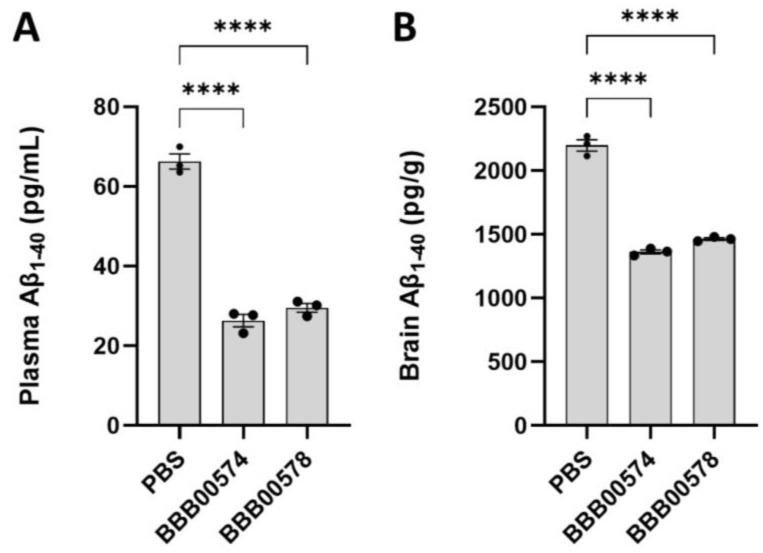
Aβ_1–40_ levels were quantified in (**A**) plasma and (**B**) brain as readout of BACE1 inhibition, in mice injected intravenously with either PBS, BBB00574, or BBB00578. The data represent means ± SEM (*n* = 3), and different conditions were compared to the PBS control group using a one-way ANOVA with Dunnett’s multiple comparisons test: **** *p* < 0.0001. Each dot represents one mouse.

## Data Availability

Data presented in this study is openly available in KU Leuven Research Data Repository (RDR) at https://doi.org/10.48804/GXJABW.

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
