# Peer review of "Novel Human/Non-Human Primate Cross-Reactive Anti-Transferrin Receptor Nanobodies for Brain Delivery of Biologics"

_pharmaceutics, 2023, doi:10.3390/pharmaceutics15061748_

Round 1
Reviewer 1 Report (Previous Reviewer 2)
The authors have effectively addressed all points raised by the reviewers. I recommend acceptance in the present form.
Author Response
We like to thank the reviewer for his/her positive feedback.
Reviewer 2 Report (New Reviewer)
This paper describes an anti-TfR nanobody fused to an anti-BACE1 antibody to improve transferrin-receptor mediated brain entry to lower A-beta levels. While this antibody may be useful to deliver antibodies to the primate brain, there are several concerns.
Minor:
The way the first sentence of the intro is written sounds like the BBB is the only biological barrier separating the CNS from the periphery. This is not true. This sentence should be rewritten.
Line 51 of the Intro: The BBB is not the only endothelium that expresses tight junction proteins. Other endothelia also express tight junction proteins which result in varying degrees of permeability depending on the perfused organ.
Major:
The data would be more accurate and easier to interpret if the nanobodies were compared to 1A11AM injection as a control rather than PBS.
Measuring brain levels of A-beta is an indirect method of measuring antibody delivery to the brain. While brain A-beta levels are important to measure, there are studies showing that lowering A-beta levels in the periphery of mice will also lower brain A-beta levels (i.e. peripheral sink hypothesis). The data would be more convincing of brain delivery if actual levels of the nanobodies fused to 1A11AM were measured rather than just A-beta levels.
Author Response
Many thanks for your time to review our manuscript, please find a response to the raised issues in attach.
Kind regards,
Maarten

Reviewer 3 Report (New Reviewer)
Summary:
Laura et al. describes the development of two nanobodies that can bind both human and cynomolgus TfR with high affinity. When tested in mice, the nanobodies were able to increase brain permeability and reduce brain Aβ1-40 levels by 40%. These findings suggest that these nanobodies could be used clinically to increase the brain permeability of therapeutic biologicals for the treatment of neurological diseases.
General comments:
In general, authors introduced 2 TfR nanobodies that bind to both human and cynomolgus TfR. This provides a potential useful tool for brain drug delivery. However, more evidence as well as the rationale behind experimental design would need to be provided to back the authors’ claim for the new nanobodies and to be considered for publication in Pharmaceutics.
Specific Comments:
For the selection process the authors described, the authors should provide more details on both the process and results.
For all in vitro assays shown in the paper, the authors should also provide sample images showing the improvement shown in the plot.
For the claim of binding to cynomolgus and the claim that this is more clinically relevant, authors would need to show the nanobodies also enhance their CNS enrichment in vivo. It could be done in a mouse with cynomolgus TfR or a cynomolgus monkey.
Author Response
Many thanks for your time to review our manuscript, please find a response to the raised issues in attach.
Kind regards,
Maarten

Round 2
Reviewer 2 Report (New Reviewer)
Please explain in text why antibody levels were not measured as you did in your reply to reviewers.
Author Response
Please see attachment

Reviewer 3 Report (New Reviewer)
Thanks for the effort in preparing the revised version. However, most of the responses didn't resolve the concerns and the authors would need to provide more effidence for their claim.
For example, for the claimed improvement, the biochemical binding assay is the only in vitro assay. Assay such as cell infectivity would be needed to show the efficient internalization of the TFr antibody.
For in vivo assay, no assay were used to show the concentration and kinetic of the antibody in vivo. The only result shown is the biomarker output.
Author Response
Dear Review,
Many thanks for your review and valuable comments. Indeed, the requested extra experiments would be valuable and should be done in case the manuscript would be a full length article. In this case however, we're sharing a short communication and hence we've only performed the key in vivo experiments needed. Therefore, in consultation with the editor, we decided not to perform the requested extra experiments.
Kind regards,
Maarten Dewilde
This manuscript is a resubmission of an earlier submission. The following is a list of the peer review reports and author responses from that submission.
Round 1
Reviewer 1 Report
1, There are many typos and grammar mistakes all over the manuscript.
2, The authors should enhance the relationship between the nanobody and the biodistribution modulation, to increase the rationality for the submission to the journal pharmaceutics.
3, The experimental part is not convincing, where multiple in vitro and in vivo models should be used to prove the prospect. A single experiment is far from enough.
4, Positive control group is missing in Fig. 3.
5, The BBB-crossing mechanism should be studied.
6, The title is not proper, since no drugs were used.
Reviewer 2 Report
The communication entitled: “Novel human/non-human primate cross-reactive anti-transfer- 1 rin receptor nanobodies for brain drug delivery.” is promising and might advance the field. However, serious issues should be addressed before this paper can be accepted for publication as follows:
1. The authors should follow Pharmaceutics Guidelines, where all Figures should be embedded in text close to where they are mentioned.
2. Please number all sections and sub-sections.
3. Please describe all abbreviations at the first place they are mentioned in the text (For example, ELISA and others).
4. Please unify either anti-body/antibody.
5. The introduction should be improved by presenting more recent studies on improving brain delivery through nanobodies.
6. The authors did not address the critical question that arose from the title of their communication: how did the nanobodies previously discovered by the authors improve brain drug delivery? Did they use a specific drug as a model, which does not normally pass the BBB, but its passage increased upon conjugation with the discovered nanobodies? Authors should conduct such an experiment to prove their claims.
7. Nanobodies library generation should be included in the materials and methods section.
8. A materials sub-section should be introduced to describe all materials involved in this communication.
9. A statistical analysis sub-section should be introduced in the materials and methods section.
10. The authors should mention how they isolated the nanobodies involved in this work.
11. The authors should classify their experimental work into in vitro and in vivo, as it confuses the reader. For instance, Bio-layer interferometry is an in vitro technique.
12. The effect of delivery route would definitely affect brain exposure. In this regard, authors should conduct in vivo binding tests to test the ability of the nanobodies to pass the BBB using different routes of administration (ICV, IV, IP, injections)
13. The authors should describe in detail how they conducted ELISA, also, Please show the binding curves obtained from ELISA.
14. Since in vitro binding does not necessarily means in vivo brain crossing, The authors should in vivo validate the anti-TFR nanobody.
15. The discussion section is the most important part of the paper. However, this part is not well- written. The results and discussion section should be better presented to highlight the most significant and unexpected results, and identify correlations, patterns, and relationships among the data, speculations, limitations of work, and deductive arguments.Also, all results should be more integrated with the discussion and should be supported by state-of-the-art studies. Also, the findings of the antibacterial and wound healing assays (That the authors will do in revisions) should be discussed and compared to similar recent studies.
16. The conclusions section is too brief and does not highlight this communication's main findings and should be improved. Ideally, the Conclusions section reviews the key findings of your work and explains the specific ways in which this work fundamentally advances the field relative to prior literature. The Conclusion section should summarize the manuscript's results and their importance, discuss ambiguous data and recommend further research. Furthermore, an effective conclusion should offer closure to a paper, leaving the reader feeling satisfied that the concepts have been fully elucidated.